# Zero-Shot Open-Vocabulary OOD Object Detection and Grounding using Vision Language Models

Poulami Sinhamahapatra[1,2], Shirsha Bose[1,2], Karsten Roscher[1], and Stephan Günnemann[2]

[1]Fraunhofer IKS, Germany
[2]Technical University of Munich, Germany

## Abstract

Automated driving involves complex perception tasks that require a precise understanding of diverse traffic scenarios and confident navigation. Traditional data-driven algorithms trained on closed-set data often fail to generalize upon out-of-distribution (OOD) and edge cases. Recently, Large Vision Language Models (LVLMs) have shown potential in integrating the reasoning capabilities of language models to understand and reason about complex driving scenes, aiding generalization to OOD scenarios. However, grounding such OOD objects still remains a challenging task. In this work, we propose an automated framework zPROD for zero-shot promptable open vocabulary OOD object detection, segmentation, and grounding in autonomous driving. We leverage LVLMs with visual grounding capabilities, eliminating the need for lengthy text communication and providing precise indications of OOD objects in the scene or on the track of the ego-centric vehicle. We evaluate our approach on OOD datasets from existing road anomaly segmentation benchmarks such as SMIYC and Fishyscapes. Our zero-shot approach shows superior performance on RoadAnomaly and RoadObstacle and comparable results on the Fishyscapes subset as compared to supervised models and acts a baseline for future zero-shot methods based on open vocabulary OOD detection.

## 1 Introduction

With the emergence of large-scale pre-trained models or foundation models, the domain of Artificial Intelligence has experienced a huge paradigm shift. These foundation models are trained on large amounts of data across multiple different domains consisting of millions or billions of parameters over several weeks or months. The immense training scale allows such models to capture sufficient generic knowledge about the world, and thus serve as a 'foundation' to be effectively utilized for downstream tasks, often in a zero-shot manner.

The recent advancements of foundation models in the form of Large Language Models (LLMs) such as BERT[1], GPT-4 [2], Llama [3] etc have sparked

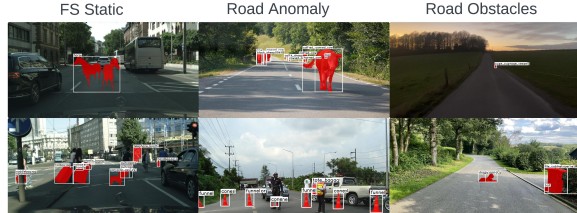

FS Static     Road Anomaly     Road Obstacles

Automated zero-shot OOD object detection, segmentation and instance prediction

**Figure 1.** Zero-shot open-vocabulary OOD object detection, segmentation and grounding using our automated framework zPROD on the test images for mentioned datasets.

development of Multimodal Large Language Models. They aim to merge the reasoning capabilities of LLMs with the rich and diverse feature space of other data modalities such as image, video, audio, and point cloud data. Different modalities provide a diverse range of tasks to be solved, which in turn enhances the overall performance in categorizing the data.

Autonomous vehicles (AV) are tasked with comprehending one of the most complex perception tasks in the form of complicated traffic scenes. Traffic scenes can be very unpredictable with a high degree of variance. Automated driving (AD) system needs to precisely perceive its surroundings, follow intricate rules, make proper decisions, and navigate with certainty. Traditional data-driven algorithms follow a modular approach that relies on three key components: perception, prediction, and planning. However, since they are trained with limited data in closed scenarios, they have been found to be inadequate in generalizing towards *open domain* like perceiving open vocabulary objects and dealing with corner cases [4]. LLMs have shown quite a promise in tackling such issues by incorporating linguistic reasoning into complex cases. However, their application has been limited to prediction and planning systems, due to their inability to process visual data. Now, with cutting-edge LVLMs such as GPT-4V [5], newer possibilities have cropped up in handling the perception component. Preliminary experiments [6] with GPT-4V on a subset of CODA dataset [4] indicate a strong grasp of traffic understanding and generalization abilities for OOD scenarios but strug-

Proceedings of the 6th Northern Lights Deep Learning Conference (NLDL), PMLR 265, 2025.

gle with vision grounding tasks such as localizing entities and specifying pixel-level coordinates. Moreover, often verbose prompt-based communication makes it difficult to derive precise instructions.

In this work, we demonstrate an automated zero-shot approach to detect and ground OOD objects in the scene using LVLMs. We leverage LVLMS capable of visual grounding and object detection such as APE [7], for tackling the problem of OOD detection as an open-vocabulary detection (OVOD) task. It strives to merge the detection, instance, and semantic segmentation tasks together with language-aided grounding. While many LVLMs exist such as Grounding-DINO [8], YOLO-world [9] etc, not every model is trained to perform the combined tasks. APE is a State-of-the-Art(SOTA) universal perception model in an instance-level sentence-object matching paradigm. We propose our method zPROD for **z**ero-shot **pr**omptable **O**OD **d**etection that requires no domain-specific fine-tuning. We propose a pipeline where the LVLM is prompted to detect all the known objects in the Operation Design Domain (ODD)/ In-domain and thus all the objects detected beyond the ODD including segmentation noise, are systematically determined whether they are potential OOD objects or not. Since we can precisely detect the OOD object, we determine whether the OOD object is on the road or the track of the ego-vehicle to further facilitate the perception and decision-making systems of AV. [10] show zero-shot inference on vision foundation models for OOD object detection. To the best of our knowledge, there exists *no-prior work or benchmark* showing zero-shot inference on LVLMs for OOD detection in an open-vocabulary setting and thus, zPROD can serve as a baseline for future work in this direction.

To summarise, the key contributions of our proposed framework zPROD are:

- zPROD is a first novel zero-shot framework for promptable open vocabulary OOD object detection, segmentation and grounding for AD application. We contribute and demonstrate two algorithms to detect plausible OOD objects to facilitate perception of AVs in this setup.

- zPROD is based on zero-shot inference on foundation models eliminating the need for tuning detector thresholds for every new dataset, thus avoiding a complex step during deployment. It relies on proposed processing steps taking into account all the instance predictions.

- We compare zPROD with existing fully supervised methods on SMIYC [11], Fishyscapes [12] benchmarks and show that zPROD outperforms them in RoadAnomaly and RoadObstacle datasets and performs comparably on Fishyscapes subsets.

## 2 Related Work

**Foundation models in Automated Driving:** Foundation models in AD leverage diverse web data and vast amounts of data generated by AVs showing superior generalization capabilities, potentially making SAE L3 [13] driving automation more realizable. In perception, LLMs can access real-time information from external APIs, such as HD maps, traffic reports, and weather updates, to enhance navigation and route planning. They enable user-centric communication, allowing users to express intentions for motion planning in everyday language. E.g., the GPT-driver [14] explains vehicle action recommendations, [15] assesses lane occupancy and safety, and "Drive as You Speak" [16] integrates advanced reasoning and language capabilities for personalized planning.

**Large Vision Language Models:** Incorporating the reasoning abilities of LLMs into vision data has led to the emergence of several LVLMs for diverse critical vision tasks such as Segment Anything Model [17] for prompt-based segmentation, DALL-E [18] for prompt-based image generation, Grounding DINO [8, 19] for visual grounding and open-vocabulary detection, and so on. In AD, SAM3D [20] combines bird-eye-view images from lidar point clouds with SAM for 3D object detection, however, the performance lags compared to SOTA so far. GTP-4V [5] also shows great potential for complex scene understanding and OOD generalization capabilities but at the same time struggles with the visual grounding of such entities. Thus, the integration of LVLMs with visual grounding capabilities seems paramount for OOD detection tasks.

## 3 Method

In this section, we provide an overview of our framework zPROD, as illustrated in Fig. 2. We first present the generic framework for detecting all the OOD objects in the scene and subsequently, present a refinement module for automatically detecting OOD objects appearing only on the track of the ego-centric AVs.

**Preliminaries - Universal visual perception LVLMs:** A LVLM is said to be capable of universal visual perception when it can perform the following tasks: (a) *Unified detection and grounding*, i.e. merging object detection and visual grounding in the form of instance level region-sentence matching, such that models can easily scale to query thousands of vocabulary concepts at a time, (b) *Unified image segmentation*, i.e. a universal model that can handle queries based on the semantic, instance and panoptic segmentation without causing mutual interference between things (foreground) and stuff (background) categories within each query. APE

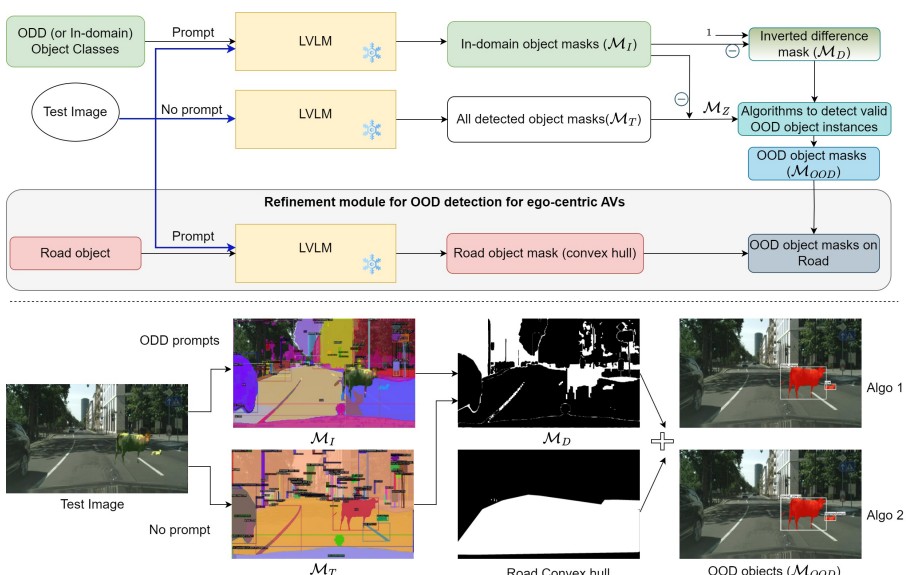

**Figure 2.** Overview of our proposed framework zPROD. Firstly, an inference with no prompts is run on frozen LVLM to detect all foreground instances($\mathcal{M}_T$) and then a second inference is run using ODD class list as prompt to obtain in-domain instances ($\mathcal{M}_I$), which are then combined into a single binary mask. The remaining non-matching instances ($\mathcal{M}_Z$) together with inverted difference mask ($\mathcal{M}_D$) are fed into our proposed Algorithms 1 and 2, to obtain final mask with OOD objects ($\mathcal{M}_{OOD}$). A refinement module is applied to detect OOD objects on the ego-track of an AV. Detected OOD objects are shown in 'red' with corresponding LVLM predictions in 'white'.

[7] trained on diverse detection, segmentation, and grounding datasets and benchmarked on over 160 datasets has proven to be robust with one suite of weights for all tasks. Thus, APE shows outstanding potential to serve as a frozen detector backbone for the task of OOD object detection due to its training with instance-level region-sentence matching in the OVOD setting.

**Framework for detecting all OOD objects:** zPROD is based on frozen pre-trained LVLMs with zero-shot detection capabilities in the OVOD setting. It depends on a few inference steps without requiring any fine-tuning on the domain dataset.

Firstly, an inference is run with the test image without any prompts, with a very low fixed LVLM detector threshold to get detections for all possible objects within the image. Assuming total $T$ instance masks are detected, collectively as $\mathcal{M}_T$. Next, the list of in-domain or ODD classes ($L$) is used as prompts to query the LVLM detector such that only object instances corresponding to ODD classes are obtained. Let there be a total of $I$ detected instance masks for ODD classes denoted by list $\mathcal{M}_I$. Lastly, remaining Z object instances ($Z = T - I$) belonging to random objects assigned to different labels than the specified object names in ODD list $L$, are collected in a list $\mathcal{M}_Z$. At this stage, all instances in $\mathcal{M}_I$ are combined into a single binary ODD mask, and this is inverted (difference with 1) to generate a *binary difference mask* ($\mathcal{M}_D$). It comprises possible unknown or OOD objects as well as noisy pixels that do not belong to any foreground

objects or remnants from imprecise segmentation, where $D = OOD + noise$.

---

**Algorithm 1:** To detect deterministic number of OOD objects

**Input:** OOD List $L =$ ['car', 'person', . . .], $N$, $\mathcal{M}_Z$, $\mathcal{M}_D$
**Output:** Mask with OOD objects $\mathcal{M}_{OOD}$

$i \leftarrow 1$, $iou \leftarrow 0$, $iou_{max} \leftarrow 0$ ;
$\mathcal{M}_{OOD} \leftarrow 0$ ;

**while** $i < N$ **do**
    **for** $1 \leq inst \leq Z$ **do**
        $iou \leftarrow \text{IoU}(\mathcal{M}_{inst}, \mathcal{M}_D)$ ;
        **if** $iou > iou_{max}$ **then**
            $iou_{max} \leftarrow iou$ ;
            **if** $pred(inst) \notin L$ **then**
                $i \leftarrow i + 1$ ;
                $\mathcal{M}_{OOD} \leftarrow \mathcal{M}_{OOD} + \mathcal{M}_{inst}$ ;
            **end**
            $\mathcal{M}_D \leftarrow \mathcal{M}_D - \mathcal{M}_{inst}$ ;
        **end**
    **end**
**end**

---

The next step is to determine valid foreground object instances amongst the noisy difference segmentation mask, which do not belong to any domain classes. This has been the most critical step in traditional object detection and segmentation tasks, as most logit-based models trained on closed-set categories struggle to discriminate between background and unknown foreground object instances. However, with foundation models trained on exhaustive classes, segmenting every foreground object has become possible without prior training on domain classes. One can query LVLMs to retrieve known objects. However, LVLMs trained on a variety of

data, tend to predict multiple object names even for similar object instances, such as a non-distinct *bird* has multiple predictions as *duck*, *seagull* etc. Also, fine-grained object instances are detected such as *shoes* which is part of *person* or *wheel* part of *car* object class. There can be several object instances that are not linguistically aligned with ODD object prompts and are falsely detected as OOD.

We provide two algorithms to determine the plausible OOD instances from the binary difference mask $\mathcal{M}_D$, as well as the remaining mask list($\mathcal{M}_Z$). In **Algorithm 1**, we assume a pre-defined maximum number of OOD objects ($N$) to be detected in the test image. Across all instances in ($\mathcal{M}_Z$) list, Intersection over Union (IoU) is checked with $\mathcal{M}_D$ and instances with maximum IoU are retained in the given iteration and removed from $\mathcal{M}_D$ for the next iteration until all $N$ OOD objects are found and appended to final OOD mask $\mathcal{M}_{OOD}$. However, in a real-time scenario, it could be impractical to estimate a pre-determined number of OOD objects. Thus, we provide **Algorithm 2** where the number of OOD objects to be detected is not deterministic. Here, across all $Z$ instances, *Normalised Intersection Score* (NIS) is calculated, i.e. for each instance mask $\mathcal{M}_{inst}$ and $\mathcal{M}_D$, where the total number of intersecting pixels are normalized only to the size of instance *inst*. This is done to detect very small OOD objects, which might otherwise be lost in the union of two masks. It is intuitive to have a NIS close to 1 for the exact matching object instance with $\mathcal{M}_{OOD}$, thus a generic NIS threshold (0.8) is fixed for all datasets. Finally, all objects in $Z$ list having NIS greater than this default threshold are considered to be OOD objects. While this might lead to false positives, in safety-critical systems, *detection of one OOD object might itself suffice to trigger changes in decision-making.*

---

**Algorithm 2:** To detect all possible OOD objects

**Input:** OOD List $L$ = ['car', 'person', ...], threshold, $\mathcal{M}_Z, \mathcal{M}_D$
**Output:** Mask with OOD objects $\mathcal{M}_{OOD}$

$val_{intersect} \leftarrow 0$ ;
$\mathcal{M}_{OOD} \leftarrow 0$ ;

**for** $1 \leq inst \leq Z$ **do**
    $val_{intersect} \leftarrow$
      Normalised Intersection($\mathcal{M}_{inst}, \mathcal{M}_D$) ;
    **if** $val_{intersect} > threshold$ **then**
        $\mathcal{M}_{OOD} \leftarrow \mathcal{M}_{OOD} + \mathcal{M}_{inst}$ ;
    **end**
**end**

---

**Refinement module to detect OOD objects on the ego-track of AVs:** As discussed above, the difference mask $\mathcal{M}_D$ can comprise a lot of irrelevant objects or noisy pixels leading false OOD detections. Different applications have different regions of relevance, e.g. AVs might only need to

consider objects lying on the ego-track such as 'road' in the road-driving scene. Thus, we propose a refinement module, where the segmentation mask for the ego-track is extracted by only prompting, say 'road', to the LVLM detector. A convex hull is extracted for all the instances corresponding to the ego-track. Finally, those OOD object masks from $\mathcal{M}_{OOD}$ are retained, which show positive IoU with the extracted convex hull of the ego-track.

Thus, using our above-proposed framework zPROD, AVs can easily perceive and trigger warnings due to the presence of OOD object instances without relying on cumbersome query-based linguistic user interactions to understand the surroundings.

# 4 Implementation Details

**Experimental Setup:** Our approach is completely based on *zero-shot inference* on frozen features of any LVLM capable of OVOD. The LVLM used in this work, is APE (D)[7] having VIT-L backbone. Using a validation set, we calculated an optimum number of OOD instances that should be detected for each dataset using Algo 1 as shown in Fig A.2 in Suppl. Similarly, for Algo 2, a generic NIS threshold for all datasets is set at 0.8 to determine plausible objects from $\mathcal{M}_{OOD}$, as described in Sec. 3.

**Datasets:** We evaluate OOD object detection for AD on existing OOD detection benchmarks for road anomaly segmentation. The 19 semantic classes given in the urban road driving dataset - Cityscapes [21], have been used as the ODD or in-domain classes for AD on road. *SegmentMeIfYouCan* (SMIYC) benchmark [11] provides two OOD datasets with real OOD objects: *RoadAnomaly21* (RA) and *RoadObstacle21* (RO). We also evaluate on a subset, *FS Static*, provided by Fishyscapes [12] (FS) benchmark It consists of generic objects taken from PASCAL VOC [22] *synthetically* overlayed on Cityscapes.

**Evaluation Metric:** While dealing with zero-shot inference on frozen foundation models, we need to detect every foreground object for OOD detection which requires the detector threshold to be *fixed* at a low value. *The zero-shot setup is different from traditional supervised methods* which depended upon logits predicted over closed-set categories based on varying detector thresholds and thus the corresponding benchmark metrics in the SMIYC[11], Fishyscapes[12] are no longer relevant. Hence, we report Intersection over Union (IoU) and mean F1 metrics with a fixed minimum detector threshold and compare with methods that report these metrics. More details on datasets and evaluation in Sec. A.1 in Supplementary.

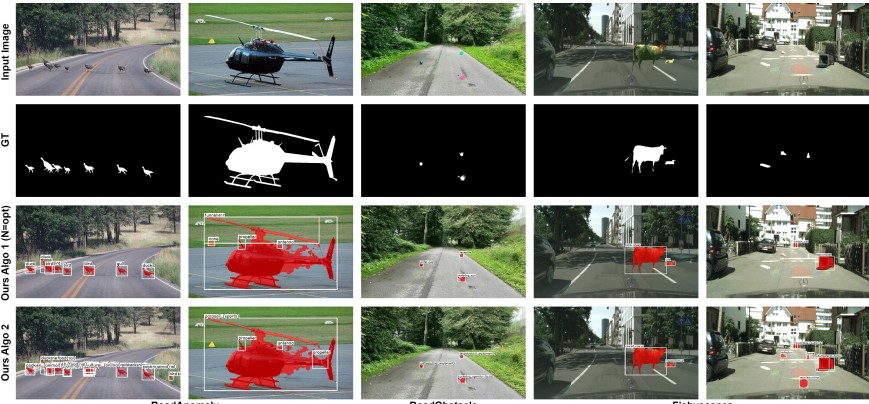

**Figure 3.** Qualitative results using zPROD for zero-shot open vocabulary automated OOD object detection and segmentation across multiple datasets for the automated driving scene. Comparison results with our proposed algorithms to detect and predict OOD object classes on the road - for pre-determined optimum number (Algo 1) and all possible number (Algo 2) of OOD objects. Detected OOD object instances are shown in red and corresponding predictions in white.

# 5 Results and Discussion

In this section, we present experiments using zPROD for showing zero-shot promptable OOD detection and grounding in AD. Table 1 shows comparison of our zero-shot method zPROD with SOTA-supervised methods reporting IoU and mean F1 metrics by [23], and qualitative samples for comparison with SOTA are presented in Fig. A.3 in Supplementary.

To the best of our knowledge, *there exists no prior work using LVLMs for zero-shot OOD object detection, segmentation or grounding in AD benchmarks or otherwise.* Thus, there exists no direct baselines or established evaluation metrics for such zero-shot methods. It should be noted that all other supervised methods are explicitly trained on domain data (ODD) object classes for pixel-wise segmentation. Thus, we show our baseline results for zero-shot OOD object detection using straight-forward approach of prompting the ODD objects and removing them. Further, in order to obtain refined results, we present results with our zPROD for both Algorithms: **Algo 1** for a deterministic number of OOD objects ($N$), and **Algo 2** for all possible OOD objects independent of $N$. For Algo 1, an optimum value of $N$ is derived from the average number of OOD objects appearing in each OOD dataset using a held-out validation set given in Fig. A.2.

In Table 1, we observe that the baseline results from directly using the difference mask $M_D$ in Fig. 2 are quite sub-optimal across all datasets since $M_D$ is quite noisy, particularly for datasets with very small OOD objects like in RO. This necessitates the need of our proposed algorithms used in zPROD for deducing the precise OOD objects. In RA and RO datasets, zPROD combined with algorithms outperform SOTA methods in all three cases via significant margin in both IoU and mean F1. While

| Method | RoadAnomaly | | RoadObstacle | | FS Static | |
|---|---|---|---|---|---|---|
| | IoU | mean F1 | IoU | mean F1 | IoU | mean F1 |
| *Trained on domain data* | | | | | | |
| DenseHybrid [24] | 26.51 | 21.34 | - | - | 23.54 | 11.15 |
| Synboost [25] | 27.22 | 29.93 | | | 32.81 | 25.67 |
| PEBAL [26] | 33.8 | 23.87 | - | | 26.92 | 13.31 |
| RPL+CoroCL [27] | 50.97 | 24.64 | - | - | 36.46 | 13.16 |
| S2M [23] | **58.49** | **61.66** | - | - | **69.99** | **70.24** |
| *Zero-shot inference using Foundation models* | | | | | | |
| Baseline | 14.93 | 21.49 | 0.7 | 1.39 | 3.85/ 5.77* | 6.94/ 10.41* |
| zPROD Algo 1 (N=1) | 65.68 | 69.44 | 39.72 | 43.48 | 45.54/ 68* | 46.92/ 66* |
| zPROD Algo 1 (N=opt) | **89.53** | **94.35** | 41.15 | 48.72 | **46.76/ 70.2*** | **48.29/ 72*** |
| zPROD Algo 2 | 63.37 | 66.4 | **43.07** | **49.85** | 38.08/ 53.75* | 40.95/ 59.17* |

**Table 1.** Results on RoadAnomaly, RoadObstacles, FS Static and FS Lost&Found. We separate the methods based on whether they were trained specifically with supervision on domain data (Cityscapes) or they were directly used for zero-shot inference from pre-trained foundation models. The best results are marked in bold in each category. * indicates results where In-domain object categories appearing as OOD are removed from test data. We observe zPROD algorithms outperform supervised methods in RA and RO datasets as well as FS Static when dubious images are removed.

OOD objects are relatively bigger in RA as compared to very small objects in RO, leading to overall worse performance in the latter, however, no results are reported by existing SOTA methods using these metrics. We are *one of the first to show reasonable performance on RO* due to the strong generalization ability of LVLM (here, APE) to detect and ground diverse foreground instances. In FS Static, some of the synthetic overlayed OOD objects belong to domain classes (ODD) shown in Fig A.1 in Suppl. Surprisingly, most supervised methods continue to detect these domain objects as OOD as they are probably susceptible to change in texture, however using LVLMs they are correctly grounded to domain class, although leading to reduced performance as no OOD is detected. Thus, we also report results

by excluding such dubious images.

In Fig. 3, we show examples with our algorithms across OOD datasets. For detecting multiple OOD objects, we find optimal results with $N = opt$ using Algo 1 as well as Algo 2, such as all the birds in Col. 1 or the very small objects like *watering can, water jug* in Col. 3. Since Algo 2 is independent of $N$, we find in Col. 5, it detects all possible OOD objects in the scene beyond the annotated OOD in GT, such as *dumpster, postbox* etc. This might lead to false positives during evaluation, however for practical purposes Algo 2 helps in detecting more OOD objects in a test image than originally annotated. However, often multiple boxes with closely matching semantic object categories are predicted for instances in Algo 2, based on multiple predictions generated by the LVLM for the same object. In such cases, the maximum IoU matching provided by Algo 1 is able to provide the most confident prediction for each instance.

## 6 Conclusion

In this work, we proposed zPROD - a novel automated zero-shot promptable method to detect open-vocabulary OOD objects using frozen LVLMs capable of visual grounding. It aims to detect and ground OOD objects in AD application, which is often difficult for supervised methods trained on closed-set categories. We proposed two algorithms, one based on detecting the pre-determined number of OOD objects and the other for detecting all possible OOD for practical use-cases. In future, this approach can also be used for creating datasets with real OOD objects from unlabelled data, for future benchmarks without much added annotation effort. Lastly, we believe this framework can serve as a useful first baseline for future zero-shot methods on foundation models in the open vocabulary setting for OOD detection.

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

# A  Appendix

## A.1  Dataset and Evaluation

**Dataset:** SegmentMeIfYouCan (SMIYC) benchmark [11] provides two OOD datasets with real OOD objects: RoadAnomaly21 (RA) and RoadObstacle21 (RO). RA consists of 100 test and 10 validation images with real objects or animals as OOD appearing anywhere in the scene. In contrast, RO has OOD objects appearing on the road/ ego-track. SMIYC benchmark withholds the GT for

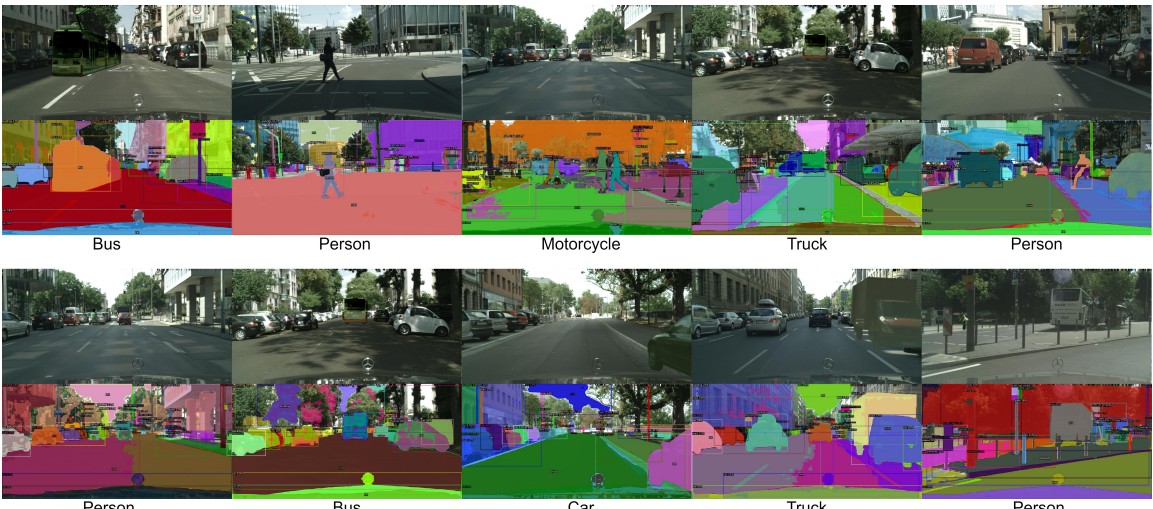

**Figure A.1.** Sample images from FS Static dataset which have annotated OOD objects which actually belong to In-domain or ODD class list of Cityscapes dataset. zPROD using inference on frozen LVLM correctly shows instance predictions. The captions for each sample indicate the OOD object annotated and correctly predicted, where each of these specific OOD object instances actually belongs to the Cityscapes In-domain class.

the test set, where the scores are only accessible by submitting the method to the official benchmark.

We also evaluate on a subset, FS Static, provided by Fishyscapes [12] (FS) . As pointed out in Sec. 5, we note that 10 images from the FS Static test set have falsely annotated OOD objects whose classes overlap with In-domain classes given in Cityscapes [21] such as *bus, truck, car, person,* etc as shown in Fig. A.1. Surprisingly, all the SOTA supervised methods continue to detect these domain objects as OOD as probably they detect the change of texture for the pasted synthetic OOD object, rather than object itself. However, LVLMs such as APE are able to correctly detect and ground them to respective ODD classes. Thus, this leads to reduced performance as there are no OOD objects detected where there are some annotated. We verify this by observing an increase in performance with our zPROD when such dubious images are eliminated.

**Evaluation:** SMIYC and Fishyscapes are road anomaly segmentation benchmarks that support supervised methods and metrics explicitly trained on domain data. These benchmarks require logits varied over confidence-based detector thresholds over closed-set classes and thus, are incompatible with methods such as ours based on zero-shot inference on foundational models trained on a large number of classes. Thus, the evaluation metrics on these benchmarks such as Average Precision (AP), False Positive Rate (FPR), etc which require variation over confidence thresholds, are no longer relevant with zero-shot methods which do not require choosing any threshold for every dataset. We show evaluation on all predicted instances from the frozen LVLMS using Intersection over Union (IoU) and F1. Thus, we are unable to make benchmark submissions. Since

the SMIYC test setup withholds the GT for test data, we are forced to show these evaluations on the validation split (where GT is given) for RA and RO and show visibly impressive qualitative result images from the test data in Fig. 1 and A.3.

## A.2  Additional Results

**Ablation to determine the optimum number of OOD objects:**

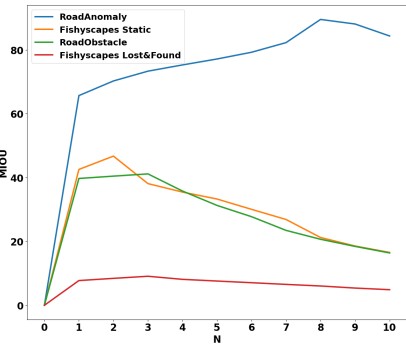

**Figure A.2.** Ablation study to determine $N = opt$ for Algo 1 for respective datasets by observing IoU values over a range of N.

In Sec. 3, we proposed Algo 1 which aimed to detect pre-determine the number of OOD objects for each dataset. Although we argue that determining only one OOD object is sufficient to trigger warnings in safety-critical systems such as in AD, however for the sake of completeness and comparison with other supervised methods, we deduce the optimum number of OOD objects ($N = opt$) appearing in each of the given datasets. We conducted an ablation study of IoU on a held-out validation dataset splits, over a

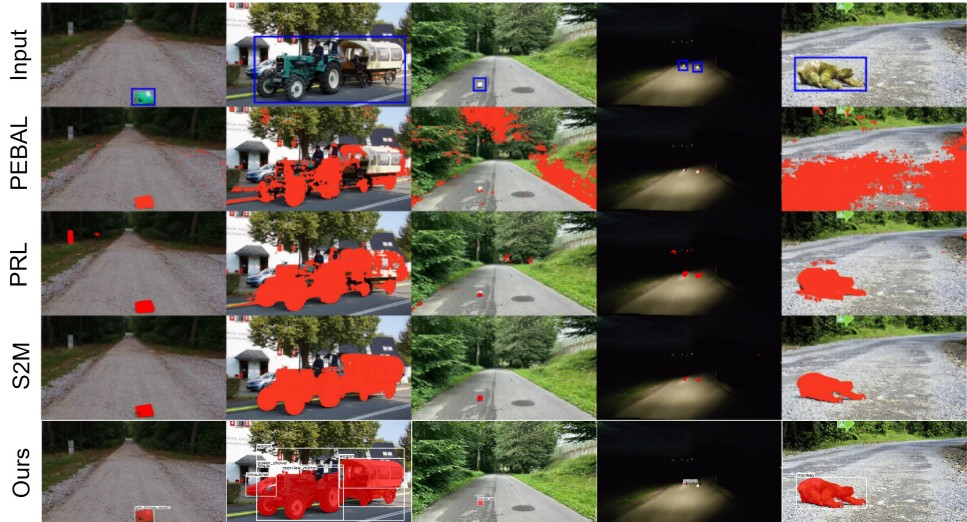

**Figure A.3.** Qualitative results for zero-shot open-vocabulary OOD object detection, segmentation and grounding using our zPROD on test images from RA and RO datasets in comparison to some SOTA supervised methods as produced by [23]. It shows impressive performance on a diverse range of OOD objects and scenes along with instance prediction. The 'blue' box on the input images denotes the OOD object. Detected OOD object instances are shown in 'red' and corresponding predictions in 'white'.

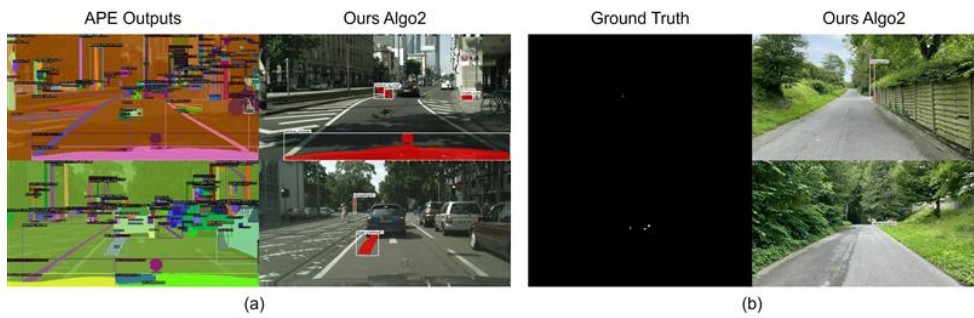

**Figure A.4.** Examples showing potential failure cases - (a)In-domain objects such as *car* predicted as similar object category like *police cruiser* (top) and *pole* as *street light* (bottom) and thus, falsely detected as OOD; (b) OOD objects are very small and match with the background, thus often fail to get detected.

range of values of N, shown in Fig. A.2. $N = opt$ was chosen at the value of N with maximum IoU.

**Qualitative comparison with SOTA:** In Fig. A.3, we show qualitative results on test images from RA and RO datasets in comparison to SOTA-supervised methods. We note that the supervised SOTA methods were not implemented but rather directly adapted from the results provided by S2M[23]. We show as compared to methods relying on supervised training, our zero-shot method provides visibly impressive performance along with grounded instance predictions. It demonstrated generalisability over a diverse range of OOD dataset variations such as gravel roads, forest roads, unknown vehicles such as carts, small objects lying on a road, night-time scenes as well as unknown animals.

## A.3 Discussion on current limitations

As shown in Fig. A.4, in the following two scenarios sometimes failure cases are obtained. Firstly, similar object categories are often predicted with different names by APE, such as *car* from In-domain object category often predicted as *vehicle, automobile, police cruiser* which do not match with prompts of In-domain and thus are falsely detected as OOD (Fig. A.4(a), top). Similarly, *pole* from In-domain category is often predicted as *street light, lamp post, electric pole* and thus get falsely detected as OOD (Fig. A.4(a), bottom). Secondly, for datasets with small objects such as RO, although zPROD reports reasonable results compared to many SOTA supervised methods that do not report results, however very small objects that match with background often fail to get detected such as the small obstacles in the Fig. A.4(b) in similar colours.

