# OpenReview forum: "Zero-Shot Open-Vocabulary OOD Object Detection and Grounding using Vision Language Models"
_NLDL.org/2025/Conference — NLDL 2025 Oral_

### Official Review · Reviewer_XbXC · 2024-09-22
**Good paper which presents a zero-shot framework for OOD object detection**

**Confidence:** 3

**Summary:**

SUMMARY: This paper presents two zero-shot algorithms for out-of-distribution (OOD) object detection using Large Vision Language Models (LVLMs). Experiment shows that the proposed method outperforms its competitors.

**Strengths:**

1. The paper addresses OOD object detection, which is an important problem in computer vision
2. The proposed methods are zero-shot
3. The experiment on standard datasets showed the superior performance of proposed methods.

**Weaknesses:**

1. The proposed methods are still simple

**Final Rebuttal Confidence:**

3

**Final Rebuttal Justification:**

I would like to keep my rating

**Justification:**

The proposed methods levarage LVLMs, enabling a zero-shot framework for OOD object detection. The experiment on standard benchmarks showed a superior performance of proposed methods. Although two proposed algorithms are still simple, their strengths outweights weaknesses.

---

### Official Review · Reviewer_irU7 · 2024-09-29

**Confidence:** 4

**Summary:**

The work proposes a zero-shot framework for OOD object detection and grounding in an autonomous vehicle setting. The framework leverages the capabilities of frozen large vision language models (LVLM) to first detect all objects in the image. In-domain objects are then detected via prompting the LVLM with a list of in-domain classes and the list of potential OOD objects is then retrieved by subtracting the masks of in-domain objects from the mask containing all objects. As the resulting OOD mask will contain noise/artifacts, two algorithms are introduced to retrieve the final list of OOD masks. These two algorithms take as input the inverse in-domain mask and the noisy OOD mask and either rely on the assumption that the number of OOD objects is known or that a threshold has been selected that considers the overlap between instances in the two masks. Finally, the LVLM is prompted to return the road mask and object outside the convex hull of the road are removed.

**Strengths:**

- Overall the proposed method is sound and while there exist approaches that address the object grounding and detection task, the focus on OOD categories is novel.
- The paper is well written and the methodology is presented in a clear manner.
- The proposed approach is intuitive.
- The work addresses an important problem of OOD object detection for autonomous driving.

**Weaknesses:**

- In the evaluation, the authors claim that the model outperforms prior approaches on the RoadAnomaly and RoadObstacle datasets while achieving competitive results on Fishyscapes (FS). However, as no results are included for RoadObstacle for the baseline methods and the difference between the best baseline approach and the proposed method on the FS dataset is quite significant, results appear overstated. Given that the experiments compare the zero-shot approach with supervised models, the reviewer believes that the significant differences on the FS dataset are not a significant problem.

- The authors mention several places that a very low fixed LVLM detector threshold is critical, however, it is not specified what this threshold is set to.

- It would have been beneficial to include a discussion of limitations and/or failure cases.

Minor:
L298 Figure reference missing.

**Final Rebuttal Confidence:**

4

**Final Rebuttal Justification:**

After going over the rebuttal and changes that the authors have made to the document, I believe that the authors have addressed the majority of the concerns and recommend acceptance of the manuscript. The paper makes an intuitive and simple, but effective contribution and the work is presented in a clear manner. The authors should, however, add a description of the newly added baseline to the manuscript.

**Justification:**

Overall, the work proposes a solution to an interesting task. While the methodological novelty mostly consists of the application of promotable LVLMs and can thus be considered limited, the reviewer believes that the intuitive solution of this new tasks warrants acceptance. The reviewer does, however, strongly encourage the authors to revise the description of the experimental results to more accurately reflect the obtained results and not overstate them.

---

> ### Author Rebuttal · Authors · 2024-10-23
>
> We thank the reviewer for the very detailed reviews and feel encouraged that the reviewer finds our method intuitive and novel for the OOD detection task. Addressing the concerns :
>
> W1 – We sincerely apologize for the confusion with the numbers reported for the FS static results in Table 1, due to a typo with misplaced decimal. The IOU and mean F1 for FS Static (where In-domain object categories appearing as OOD are removed from test data as discussed in Sec. A.1) are observed to be $70.2$% and $72$% respectively using Algo 1, which outperforms the SOTA supervised methods. Thus, we had remarked our method to be competitive for this dataset. It is important to note that existing vision based supervised methods continue to detect In-domain objects annotated as OOD in test data as OOD objects, probably due to change of texture as synthetic object crops were pasted as OOD objects although object itself belonged to In-domain categories in many cases. However, we show in our method using VLMs, that it is no longer susceptible to such texture problems and every object is predicted their respective category. Thus, including such dubious images with In-domain objects as OOD unfairly penalizes VLM based OOD detection methods, thus we show additional evaluation removing such images and find improved results which are quite competitive. We strive to include appropriate description of our results in the final paper.
>
> Regarding RoadObstacles dataset, most supervised methods find it harder to detect such small obstacles and thus do not report results. We now included baseline results using LVLM and we find that due to very noisy difference masks ($M_D$) , it is almost impossible to directly filter the small obstacles as OOD. However, LVLMs are still able to detect some prominent tiny obstacles and thus using our algorithms, we can show fairly better performance as compared to existing methods on RoadObstacle dataset.
>
> W2 – Since our method relies on detecting every possible object instance predictions to detect possible OOD instances, thus a low detector threshold is fixed. For our experiments, the threshold was set to $0.1$ or $10$% confidence.
>
> W3 – We have now included some of the failure cases in Appendix Sec A.3 and Fig. A.4. Notably, most common failure cases arise when In-domain objects are predicted with names of closely matching semantic objects by APE such as 'poles' as 'street lights' or 'cars' as 'police cruiser' etc leading to false OOD detections. Moreover, as mentioned above, very small objects in datasets like RoadObstacles pose challenge in reliable detection.

---

### Official Review · Reviewer_a9Xt · 2024-10-03
**Review for submission #49**

**Confidence:** 4

**Summary:**

This work proposes to leverage the Large vision-language model (LVLM) to perform zero-shot out-of-domain (OOD) object detection. The authors first query all the foreground instances within the scenes and propose two algorithms to determine the plausible OOD instances. They also propose a refinement module that enhances the result by focusing on the road area. Experiments show that the proposed approach can surpass some supervised methods in RoadAnomaly.

**Strengths:**

1. I agree that the research paradigm has shifted since the emergence of the large model. Hence, the motivation of this work, using LVLM for OOD detection for AD scenarios, is clear.
2. The proposed techniques are intuitive and make sense to me.
3. Experiments show they can exceed some supervised methods by a large margin.

**Weaknesses:**

1. The proposed pipeline is a post-processing. In contrast to the comparison methods, the technical contribution may not be sufficient.
2. I think Table 1 lacks a baseline method. That is, using APE to predict the OOD objects directly.
3. I think it's better to use mathematic equations to describe the pipeline (line 199-270).

**Justification:**

1. Although I think the technical contributions of this work may not be significant, it is a first and interesting attempt to prompt the LVLM in the task of OOD detection. And the authors demonstrate competitive results, which can serve as a strong baseline for later research.
2. In general, the proposed algorithms are clear and easy to follow.

---

> ### Author Rebuttal · Authors · 2024-10-23
>
> We are encouraged by very positive, thoughtful reviews and appreciation of the novelty of this work by the reviewer. Addressing the concerns :
>
> W1 – Our method is  zero-shot and thus relies on non-trivial post-processing methodology. It shows that using large models, the OOD detection task can be much easily achieved than ever possible in the past, without spending time and resources on supervised training on specific domain data. It shows impressive generalization capabilities across multiple datasets with diverse OOD object types along with a possibility to predict the class of the OOD object.
>
> W2 – We are not sure we understand the reviewer's suggestion, so would be nice to have clarification regarding this? Given that OOD objects are supposed to be unknown and appear during test time, thus it is unclear how LVLM should be used to prompt and predict the OOD objects directly without knowing them before-hand.
>
> Nonetheless, we now included baseline results in Table 1 in the revised paper. We use the difference mask $M_D$  (See L207-219 and Fig. 2) as the baseline for OOD object detection using LVLM. For this, all the known objects are queried into the LVLM and the predicted masks are then  combined into a single binary mask and the difference from $1$ (essentially the original image) is obtained as the binary mask $M_D$. Using $M_D$, we evaluate the results for OOD detection on different datasets. We observe significantly poor results as compared to the results using our algorithms. This is because $M_D$, not only contains potential OOD objects but also a lot of noise from segmentation errors and even some background pixels which are otherwise not detected as foreground objects. These masks lead to even worse results for datasets with small obstacles such as RoadObstacle, where the ratio of false positive (noisy) pixels are much higher compared to true positives. This aptly justifies the need of our two proposed algorithms  presented as the subsequent steps of our pipeline as shown in Fig. 2, starting with $M_D$ as input.
>
> W3 – L199-240 involve pre-processing steps without much mathematical calculations, hence we illustrated this using Fig. 2.  For L240-270, we describe our algorithms which include mathematical equations and are already shown in an algorithmic manner in Algorithm 1 and 2.

---

### Meta-Review · Area_Chair_733k · 2024-11-01

**Recommendation:** Accept (Poster)
**Confidence:** 5

**Metareview:**

This paper suggests to rely on large vision-language model (LVLM) to perform zero-shot out-of-domain (OOD) object detection.

All reviewers agreed about the importance of the problem, the relevance of the techniques involved, the clarity of the paper, and the significance (to some extent) of the experimental results. However, a few negative aspects were also identified, lack of technical contributions, overstated results, lack of technical details. The authors provide a systematic rebuttal to reviewers' comments and their replies were judged valuable.

Overall, the paper brings an intuitive and simple, but effective contribution and the work is presented in a clear manner. The authors should, however, be reminded to add a description of the newly added baseline to the manuscript.

All reviewers suggest to accept the paper. Given the strength of the contribution, a poster presentation seems more appropriate.

**Suggested Changes To The Recommendation:**

3: I agree that the recommendation could be moved up

---

### Decision · Program_Chairs · 2024-11-06

**Decision:**

Accept (Oral)

**Comment:**

Given the AC positive recommendation, we recommend an oral and a poster presentation given the AC and reviewers recommendations.